# A Model to Calculate the Current–Temperature Relationship of Insulated and Jacketed Cables

**DOI:** 10.3390/ma15196814

**Published:** 2022-09-30

**Authors:** Jordi-Roger Riba, Jordi Llauradó

**Affiliations:** Electrical Engineering Department, Universitat Politècnica de Catalunya, Rambla Sant Nebridi 22, 08222 Terrassa, Spain

**Keywords:** insulated cable, polymeric insulation, cable model, temperature rise, simulation, finite difference method

## Abstract

This paper proposes and validates using experimental data a dynamic model to determine the current–temperature relationship of insulated and jacketed cables in air. The model includes the conductor core, the inner insulation layer, the outer insulating and protective jacket and the air surrounding the cable. To increase its accuracy, the model takes into account the different materials of the cable (conductor, polymeric insulation and jacket) and also considers the temperature dependence of the physical properties, such as electrical resistivity, heat capacity and thermal conductivity. The model discretizes the cable in the radial direction and applies the finite difference method (FDM) to determine the evolution over time of the temperatures of all nodal elements from the temperatures of the two contiguous nodes on the left and right sides. This formulation results in a tri-diagonal matrix, which is solved using the tri-diagonal matrix algorithm (TDMA). Experimental temperature rise tests at different current levels are carried out to validate the proposed model. This model can be used to simulate the temperature rise of the cable when the applied current and ambient temperature are known, even under short-circuit conditions or under changing applied currents or ambient temperatures.

## 1. Introduction

The demand for electrical energy is currently growing worldwide [1,2,3], so power systems load levels are increasing. It is of paramount importance to ensure that power cables operate within their thermal limits to not compromise their safe operation. The ampacity of insulated cables can be calculated by applying the method detailed in the IEC 60287 standard [4]. However, this standard only provides formulas to determine the current rating or maximum permissible current under steady-state conditions and a maximum temperature increase, but it does not develop the heat transfer equation. The same applies for the IEC 60853 [5], which develops methods for determining the cyclic and emergency current ratings of power cables, but does not provide a method for determining their temperature evolution. The IEC 60986 [6] standard, which is related to the short-circuit temperature limits of insulated cables, provides tables with the maximum permissible short-circuit temperatures to limit the *I*^2^*t* heating based on the consideration of the range of limits used by various authorities, but it does not allow the temperature evolution of the cable to be determined. The IEC 60986 recognizes that the values in the tables are safe, but they are not necessarily ideal because there are very little available experimental data on actual cables. Given the limitations of the aforementioned standards, this work contributes to this field, since it develops and validates with experimental tests a dynamic model to determine the current–temperature relationship between insulated and jacketed cables in air. The proposed model includes the core, the inner insulation layer, the outer protective jacket and the air surrounding the cable. The model considers the different cable materials (conductor, insulation and jacket) and the temperature dependencies of their physical properties.

Current rating calculations on power cables require determining the temperature of the different cable layers for a specified current or determining permissible current for a specified cable temperature [7]. Therefore, in order to perform these calculations, it is necessary to determine the heat generated due to the Joule effect within the conductor and the rate of its dissipation away from the cable, which depends on the current level, cable size, composition and laying method. To this end, the heat conduction equation must be solved by applying numerical approaches [7].

Two- and three-dimensional finite element analysis (FEA) approaches have been widely used to address this problem [8,9,10,11], because FEA simulations are widely accepted as a powerful and realistic approach to determine the electromagnetic and thermal performance of power cables and other devices intended for power systems [12]. For example, the IEC TR 62095 standard [7] suggests applying FEA methods, when the methods discussed in the IEC 60287 (steady-state conditions) and IEC 60853 (cyclic conditions) cannot be applied. However, FEA approaches involve preparing the geometry and mesh of the problem. They are often memory-intensive and time-consuming due to the number of discretized elements they require, especially when solving coupled multiphysics problems. FEA approaches also require the purchase of specific software and the periodic maintenance of expensive licenses, as well as the involvement of qualified technicians [13].

In [14], a method to calculate the temperature rise of cable systems under steady-state conditions is presented, which the authors recognize as a simple procedure compared to the transient case. In [15], an empirical transient cable model is presented, which neglects radiation losses and changes in conductor resistance, among others. In [16], a thermal model of bare and insulated conductors is presented, but it does not include the jacket. In [17], a lumped parameter thermal network for direct current medium voltage cables is presented. Although this model allows the temperature rise of the cable to be determined, its accuracy is limited, in part because of the poor level of discretization, and in part, because it neglects the temperature dependency of the main parameters of the cable (thermal conductivity, specific heat capacity and electric resistivity). Similar approaches based on lumped parameter thermal networks are presented in [18,19,20,21], with similar limitations. In [22], an approach to calculate the transient temperature of a single-core insulated cable using an analytic approach is presented, but the model does not include the jacket and different simplifications are performed in order to solve the differential equations arising from the model.

Due to the limitations of FEM-based methods (geometry preparation, meshing, computational burden or software licenses among others) or the limitations of lumped parameter methods (poor discretization or not considering the temperature dependency of the cable parameters), to overcome these drawbacks, it is highly appealing to develop fast and accurate transient models [23], if possible based on model reduction methods [24].

This paper describes a numerical method developed to determine the radial temperature distribution (from the center of the core to the outer surface) of stranded insulated and jacketed cables, which is affected by the steady or time-varying electrical current flowing through the cable and the ambient temperature. The proposed model considers the temperature dependency of the main cable parameters, such as the electric resistivity of the conductor and the volumetric densities, thermal conductivities and specific heat capacities of the different cable materials. The method proposed in this paper may be used to determine the radial temperature distribution when the current is known or to determine the temperature of the cable yielding a maximum allowable temperature. Therefore, it allows the current corresponding to pre-established temperature limits of the cable to be determined, since it solves the non-stationary heat transfer equation. The proposed model also allows steady-state, transient and dynamic problems to be solved. The steady-state problem is applied when the electric current and the ambient and cable temperatures are a constant value and independent of time. The transient problem occurs when the ambient temperature remains constant but the current undergoes a step change, thus affecting the radial temperatures, following an exponential change. Finally, the dynamic problem can be applied when the ambient temperature and/or current change over time follows any pattern. Due to the increase in load levels, cables are often pushed to their limits. The model developed in this paper can also be a useful tool for cable maintenance tasks, which can also disclose the relationship between the current, thickness and material of the insulation and jacket layers, ambient temperature and cable temperature. Finally, the comprehensive method here proposed, which fully develops the physical equations that govern the heat transfer problem, can be easily adapted to other cable configurations.

## 2. Transient Thermal Model of the Insulated and Jacketed Cable

Figure 1 shows a cross section of the cable, which includes the conductor core (orange color), the inner insulation layer (blue color) and the outer insulating and protecting jacket (gray color). It is noted that both the inner insulation and the outer jacket are layers of the cable that protect the conductor. Whereas the insulation layer isolates the current flow, the jacket is the outermost layer, which protects the conductor core and the insulation from chemical deterioration and external elements.

The temperature of the cable mainly depends on two factors, that is, the self-generated heating due to the Joule effect and the ambient temperature. The electric current is confined within the metallic conductor core, which is usually made of copper or aluminum strands, so the heat flows from the conductor to the air passing through the inner insulation layer and the outer jacket to the surrounding air.

The transient thermal behavior of bare conductors, that is, without insulation and jacket layers, can be modeled by applying the methods found in the CIGRE [25] and IEEE Std. 738 [26] standards, which state that for bare conductors, the heat balance equation results from the balance between the heat gain and the heat loss terms as,
(1)$$I_{RMS}^{2}r(T)=P_{c}+P_{r}-P_{s}+mc_{p}(T)\frac{dT}{dt}\;[W/m]$$
where *I_RMS_* (A) is the root mean square value of the current flowing through the conductor, *r* (Ω/m) is the per unit length electric resistance of the conductor, *P_c_* (W/m) and *P_r_* (W/m) are, respectively, the per unit length convection and radiation heat loss terms, *P_s_* (W/m) is the term due to the per unit length solar heat gain, *m* (kg/m) the per unit length conductor mass, *c_p_(T)* (J/(kgK)) the specific heat capacity of copper or aluminum (conductor material) and *T* (K) the mean temperature of the conductor and *t* (s) the time. Since in this paper the tests are performed indoors, the solar heat gain term *P_s_* is not considered.

It is worth noting that the resistance per unit length depends on the temperature as,
(2)$$r(T)=r_{T_{0}}[1+\alpha (T-T_{0})]$$
where *T*_0_ (K) is the reference temperature (usually 20 °C or 293.15 K), *T* (K) is the mean temperature of the analyzed conductor node and α (K^−1^) is the temperature coefficient of the resistance. It is known that with ac supply the current density throughout the cross-section of the conductor cannot be uniform due to the eddy current effects [27], but the measured value of $r_{T_{0}}$ already includes such effects.

When dealing with insulated and jacketed cables, the analysis is more complex than that shown in (1), so a more detailed study is required, which is detailed in the next sections. In this case, three different materials are involved in the three layers of the cable, for example, copper in the conductor core, XLPE in the inner insulation layer and PVC in the outer jacket. For more accuracy, the temperature dependencies of the specific heat capacities and the thermal conductivities of such materials are considered.

Figure 2 shows the temperature dependency of the specific heat of copper [28], XLPE [29] and PVC [30], while Figure 3 shows the temperature dependency of the thermal conductivity of copper [31], XLPE [32] and PVC [33]. It is worth noting that these properties are needed to solve the heat transfer equations.

### 2.1. Domain Discretization and TDMA Formulation

As explained, the temperature of an insulated cable depends on two main factors, which are the self-generated Joule (*I*^2^*r*) heating and the externally ambient temperature. Thus, in general, heat flows radially from the central part of the conductor to the outer part of the jacket. The whole domain of the cable is discretized into many discrete elements along the radial dimension of the cable, as shown in Figure 4.

The heat transfer problem is solved along the radial axis because the heat is conducted through this axis. This approach assumes that the central point of any node has the mean temperature of the discrete element where it is placed. This assumption is accurate since a small spatial step Δ*x* is chosen. The finite difference method (FDM) determines the nodal temperature from the temperatures of the two contiguous left-hand and right-hand side nodes. A tri-diagonal matrix describing the nodal temperature arises, so the temperatures can be determined by applying the tri-diagonal matrix algorithm (TDMA) [34], which has the form [12],
(3)$$a_{i}^{j}T_{{}_{,}i-1}^{j+1}+b_{i}^{j}T_{{}_{,}i}^{j+1}+c_{i}^{j}T_{{}_{,}i+1}^{j+1}-d_{i}^{j}=0$$
*i* and *j* being, respectively, the indices related to the spatial and temporal steps, so that *T_i_^j^* = *T*(*i*Δ*x*,*j*Δ*t*) corresponds to the average temperature of the *i*-th node calculated at the *j*-th time step. Finally, *a_i_^j^*, *b_i_^j^*, c*_i_^j^* and *d_i_^j^* are constant coefficients. Equation (4) calculates the nodal temperatures from the temperatures of the neighboring right-hand side and left-hand side nodes.



(4)
$$\left[\begin{array}{ccccc} b_{1}^{j} & c_{1}^{j} & & & \\ a_{2}^{j} & b_{2}^{j} & c_{2}^{j} & . & \\ & a_{3}^{j} & b_{3}^{j} & c_{3}^{j} & .\\ c_{tot}^{j} & & & & a_{tot}^{j} \end{array}\begin{array}{c} a_{1}^{j}\\ c_{tot-1}^{j}\\ b_{tot}^{j} \end{array}\right]\cdot \left[\begin{array}{c} T_{1}^{j+1}\\ T_{2}^{j+1}\\ T_{3}^{j+1}\\ .\\ T_{tot}^{j+1} \end{array}\right]=\left[\begin{array}{c} d_{1}^{j}\\ d_{2}^{j}\\ d_{3}^{j}\\ .\\ d_{tot}^{j} \end{array}\right]\;\;\mathrm{with}\;a_{1}^{j}=c_{tot}^{j}=0$$



### 2.2. Conductor Discretization

The conductor part (orange color) designated with subscript *A* includes *N_A_* nodal elements of thickness Δ*x_A_* (m) each, as shown in Figure 5, whereas the elements of the insulation layer are shown in blue color. An additional node is placed at the boundary between the conductor and the insulation layer.

To properly distribute the electric current and the electric resistance along each division, the electric resistivity *ρ_e_* (Ω·m) and the current density *J_i_* = *I_i_/S_i_* (A/m^2^) in each element are required, *S_i_* and *I_i_* being, respectively, the cross section and the current corresponding to the *i*-th element, so the resistance per unit length of the *i*-th element is *r_i_* = *ρ_e_ /S_i_* (Ω/m).

**First node (*i* = 1) analysis.** At the first node, heat is generated by the Joule effect and conducted to the adjacent outer node (right-hand side). The discretized heat transfer equation can be expressed as,
(5)$$\rho_{A}S_{1,A}c_{p,A}\frac{T_{1}^{j+1}-T_{1}^{j}}{\Delta t}=I_{i}^{2}r_{i}(T_{1}^{j})-k_{A}\frac{T_{1}^{j+1}-T_{2}^{j+1}}{\Delta x_{A}}p_{1}\;$$
with,
(6)$$\left\{ \begin{array}{l} S_{1,A}=\pi {\left[x_{1}+0.5\Delta x_{A}\right]}^{2}\\ p_{1}=2\pi \left[x_{1}+0.5\Delta x_{A}\right] \end{array}\right.$$
where *ρ_A_* (kg/m^3^) is the mass density of the conductor material, *c_p,A_* (J/(kgK)) is the specific heat capacity of the conductor material, *r_i_* (Ω/m) is the per unit length resistance of the *i*-th element of the conductor, *k_A_* (W/(mK)) is the thermal conductivity of the conductor material, and *T_i_^j^* (K) is the temperature of the *i*-th element calculated at time *t* = *j*Δ*t* (s).

After reorganizing the terms in (5) to match with the TDMA formulation, the TDMA coefficients of the first node result in,
(7)$$\left\{ \begin{array}{l} a_{1}=0\\ b_{1}=\frac{\rho_{A}S_{1,A}c_{p,A}}{\Delta t}+k_{A}\frac{p_{1}}{\Delta x_{A}}\\ c_{1}=-k_{A}\frac{p_{1}}{\Delta x_{A}}\\ d_{1}=\frac{\rho_{A}S_{1,A}c_{p,A}T_{1}^{j}}{\Delta t}+I_{i}^{2}r_{i}(T_{1}^{j}) \end{array}\right.$$

**Generic node (*i*) analysis.** A generic conductor node includes the heat generation and the heat conduction from the inner (left) to the outer (right) element, thus resulting in,
(8)$$\rho_{A}S_{i,A}c_{p,A}\frac{T_{i}^{j+1}-T_{i}^{j}}{\Delta t}=I_{i}^{2}r_{i}(T_{i}^{j})+k_{A}\frac{T_{i-1}^{j+1}-T_{i}^{j+1}}{\Delta x_{A}}p_{i-1}-k_{A}\frac{T_{i}^{j+1}-T_{i+1}^{j+1}}{\Delta x_{A}}p_{i+1}$$
where
(9)$$\begin{array}{l} \left\{ \begin{array}{l} S_{i,A}=\pi {\left[x_{i}+0.5\Delta x_{A}\right]}^{2}-\pi {\left[x_{i}-0.5\Delta x_{A}\right]}^{2}\\ p_{i-1}=2\pi \left[x_{i}-0.5\Delta x_{A}\right]\\ p_{i+1}=2\pi \left[x_{i}+0.5\Delta x_{A}\right] \end{array}\right.\\ \mathrm{with}\;2\leq i\leq N_{A} \end{array}$$

After reorganizing the terms above to match with the TDMA formulation, the TDMA coefficients of the generic node result in,
(10)$$\begin{array}{l} \left\{ \begin{array}{l} a_{i}=-k_{A}\frac{p_{i-1}}{\Delta x_{A}}\\ b_{i}=\frac{\rho_{A}S_{i,A}c_{p,A}}{\Delta t}+k_{A}\frac{p_{i-1}}{\Delta x_{A}}+k_{A}\frac{p_{i+1}}{\Delta x_{A}}\\ c_{i}=-k_{A}\frac{p_{i+1}}{\Delta x_{A}}\\ d_{i}=\frac{\rho_{A}S_{i,A}c_{p,A}T_{i}^{j}}{\Delta t}+I_{i}^{2}r_{i}(T_{i}^{j}) \end{array}\right.\\ \mathrm{with}\;2\leq i\leq N_{A} \end{array}$$

**Boundary node (*i* = *N_A_* + 1).** In the boundary node between the conductor and the insulating layer (see Figure 5), the thermal conductivity is different on each side of the node, and the Joule effect heat generation is only in the conductor, not in the insulation.
(11)$$\rho_{A}S_{i,A}c_{p,A}\frac{T_{i}^{j+1}-T_{i}^{j}}{\Delta t}+\rho_{B}S_{i,B}c_{p,B}\frac{T_{i}^{j+1}-T_{i}^{j}}{\Delta t}=I_{i}^{2}r_{i}(T_{{}_{,}i}^{j})+k_{A}\frac{T_{i-1}^{j+1}-T_{i}^{j+1}}{\Delta x_{A}}p_{i-1}-k_{B}\frac{T_{i}^{j+1}-T_{i+1}^{j+1}}{\Delta x_{B}}p_{i+1}$$
where
(12)$$\begin{array}{l} \left\{ \begin{array}{l} S_{i,A}=\pi {\left[x_{i}\right]}^{2}-\pi {\left[x_{i}-0.5\Delta x_{A}\right]}^{2}\\ S_{i,B}=\pi {\left[x_{i}+0.5\Delta x_{B}\right]}^{2}-\pi {\left[x_{i}\right]}^{2}\\ p_{i-1}=2\pi \left[x_{i}-0.5\Delta x_{A}\right]\\ p_{i+1}=2\pi \left[x_{i}+0.5\Delta x_{B}\right] \end{array}\right.\\ \mathrm{with}\;i=N_{A}+1 \end{array}$$

In the boundary node, the resistance per unit length and current only affect the region of material A (conductor), so that the current density *J_i_* = *I_i_/S_i_* (A/m^2^) is required in each element, *S_i_* and *I_i_* being, respectively, the cross section and the current of the *i*-th element. The resistance per unit length of the *i*-th element is *r_i_* = *ρ_e_/S_i,A_* (Ω/m), where *I_i_* = *S_i,A_J*_i_.

The TDMA coefficients of the boundary node result in,
(13)$$\begin{array}{l} \left\{ \begin{array}{l} a_{i}=-k_{A}\frac{p_{i-1}}{\Delta x_{A}}\\ b_{i}=\frac{\rho_{A}S_{i,A}c_{p,A}}{\Delta t}+\frac{\rho_{B}S_{i,B}c_{p,B}}{\Delta t}+k_{A}\frac{p_{i-1}}{\Delta x_{A}}+k_{B}\frac{p_{i+1}}{\Delta x_{B}}\\ c_{i}=-k_{B}\frac{p_{i+1}}{\Delta x_{B}}\\ d_{i}=\frac{\rho_{A}S_{i,A}c_{p,A}T_{i}^{j}}{\Delta t}+\frac{\rho_{B}S_{i,B}c_{p,B}T_{i}^{j}}{\Delta t}+I_{i}^{2}r_{i}(T_{{}_{,}i}^{j}) \end{array}\right.\\ \mathrm{with}\;i=N_{A}+1 \end{array}$$

Although the conductor has already been discretized, as the thermal conductivity of copper is approximately three orders of magnitude larger compared to that of the surrounding insulating materials (see Figure 3), a single temperature value can be assigned to the conductor, as the temperature drop over it is almost negligible.

### 2.3. Inner Insulation Discretization

The elements in the inner insulation layer (blue color) are expressed with subscript *B*, which includes *N_B_* nodal elements, as shown in Figure 6, whereas the elements of the outer jacket are represented in gray color. An additional node is placed at the boundary between the insulation layer and the outer jacket.

Regarding the insulation layer, no internal heat source exists, so the temperature only varies along the radial axis [35].

**Generic node (*i*).** Since no heat is generated within the generic node of the insulation layer, the heat transfer equation results in,
(14)$$\rho_{B}S_{i,B}c_{p,B}\frac{T_{i}^{j+1}-T_{i}^{j}}{\Delta t}=k_{B}\frac{T_{i-1}^{j+1}-T_{i}^{j+1}}{\Delta x_{B}}p_{i-1}-k_{B}\frac{T_{i}^{j+1}-T_{i+1}^{j+1}}{\Delta x_{B}}p_{i+1}$$
where
(15)$$\begin{array}{l} \left\{ \begin{array}{l} S_{i,A}=\pi {\left[x_{i}+0.5\Delta x_{B}\right]}^{2}-\pi {\left[x_{i}-0.5\Delta x_{B}\right]}^{2}\\ p_{i-1}=2\pi \left[x_{i}-0.5\Delta x_{B}\right]\\ p_{i+1}=2\pi \left[x_{i}+0.5\Delta x_{B}\right] \end{array}\right.\\ \mathrm{with}\;N_{A}+2\leq i\leq N_{A}+N_{B} \end{array}$$

The TDMA coefficients of a generic node of the inner insulation layer result in,
(16)$$\begin{array}{l} \left\{ \begin{array}{l} a_{i}=-k_{B}\frac{p_{i-1}}{\Delta x_{B}}\\ b_{i}=\frac{\rho_{B}S_{i,B}c_{p,B}}{\Delta t}+k_{B}\frac{p_{i-1}}{\Delta x_{B}}+k_{B}\frac{p_{i+1}}{\Delta x_{B}}\\ c_{i}=-k_{B}\frac{p_{i+1}}{\Delta x_{B}}\\ d_{i}=\frac{\rho_{B}S_{i,B}c_{p,B}T_{i}^{j}}{\Delta t} \end{array}\right.\\ \mathrm{with}\;N_{A}+2\leq i\leq N_{A}+N_{B} \end{array}$$

**Boundary node (*i* = *N_A_* + *N_B_* + 1).** The heat transfer equation of the boundary node between the inner insulation layer and the outer jacket can be expressed as,
(17)$$\rho_{B}S_{i,B}c_{p,B}\frac{T_{i}^{j+1}-T_{i}^{j}}{\Delta t}+\rho_{C}S_{i,C}c_{p,C}\frac{T_{i}^{j+1}-T_{i}^{j}}{\Delta t}=k_{B}\frac{T_{i-1}^{j+1}-T_{i}^{j+1}}{\Delta x_{B}}p_{i-1}-k_{C}\frac{T_{i}^{j+1}-T_{i+1}^{j+1}}{\Delta x_{C}}p_{i+1}$$
where
(18)$$\begin{array}{l} \left\{ \begin{array}{l} S_{i,B}=\pi {\left[x_{i}\right]}^{2}-\pi {\left[x_{i}-0.5\Delta x_{B}\right]}^{2}\\ S_{i,C}=\pi {\left[x_{i}+0.5\Delta x_{C}\right]}^{2}-\pi {\left[x_{i}\right]}^{2}\\ p_{i-1}=2\pi \left[x_{i}-0.5\Delta x_{B}\right]\\ p_{i+1}=2\pi \left[x_{i}+0.5\Delta x_{C}\right] \end{array}\right.\\ \mathrm{with}\;i=N_{A}+N_{B}+1 \end{array}$$

The TDMA coefficients of the boundary node between the inner insulation layer and the outer jacket result in,
(19)$$\begin{array}{l} \left\{ \begin{array}{l} a_{i}=-k_{B}\frac{p_{i-1}}{\Delta x_{B}}\\ b_{i}=\frac{\rho_{B}S_{i,B}c_{p,B}}{\Delta t}+\frac{\rho_{C}S_{i,C}c_{p,C}}{\Delta t}+k_{B}\frac{p_{i-1}}{\Delta x_{B}}+k_{C}\frac{p_{i+1}}{\Delta x_{C}}\\ c_{i}=-k_{C}\frac{p_{i+1}}{\Delta x_{C}}\\ d_{i}=\frac{\rho_{B}S_{i,B}c_{p,B}T_{i}^{j}}{\Delta t}+\frac{\rho_{C}S_{i,C}c_{p,C}T_{i}^{j}}{\Delta t} \end{array}\right.\\ \mathrm{with}\;i=N_{A}+N_{B}+1 \end{array}$$

### 2.4. Outer Jacket Discretization

The elements in the outer jacket (gray color) are expressed with subscript *C*. As shown in Figure 7, it includes *N_C_* nodal elements. It can be observed that an additional node has been added to the boundary between the outer jacket and the air.

**Generic node (*i*).** Since no heat is generated in the generic node of the outer jacket, the heat transfer equation results in,
(20)$$\rho_{C}S_{i,C}c_{p,C}\frac{T_{i}^{j+1}-T_{i}^{j}}{\Delta t}=k_{C}\frac{T_{i-1}^{j+1}-T_{i}^{j+1}}{\Delta x_{C}}p_{i-1}-k_{C}\frac{T_{i}^{j+1}-T_{i+1}^{j+1}}{\Delta x_{C}}p_{i+1}$$
where
(21)$$\begin{array}{l} \left\{ \begin{array}{l} S_{i,C}=\pi {\left[x_{i}+0.5\Delta x_{C}\right]}^{2}-\pi {\left[x_{i}-0.5\Delta x_{C}\right]}^{2}\\ p_{i-1}=2\pi \left[x_{i}-0.5\Delta x_{C}\right]\\ p_{i+1}=2\pi \left[x_{i}+0.5\Delta x_{C}\right] \end{array}\right.\\ \mathrm{with}\;N_{A}+N_{B}+2\leq i\leq N_{A}+N_{B}+N_{C} \end{array}$$

The TDMA coefficients of a generic node of the outer jacket are as follows,
(22)$$\begin{array}{l} \left\{ \begin{array}{l} a_{i}=-k_{C}\frac{p_{i-1}}{\Delta x_{C}}\\ b_{i}=\frac{\rho_{C}S_{i,C}c_{p,C}}{\Delta t}+k_{C}\frac{p_{i-1}}{\Delta x_{C}}+k_{C}\frac{p_{i+1}}{\Delta x_{C}}\\ c_{i}=-k_{C}\frac{p_{i+1}}{\Delta x_{C}}\\ d_{i}=\frac{\rho_{C}S_{i,C}c_{p,C}T_{i}^{j}}{\Delta t} \end{array}\right.\\ \mathrm{with}\;N_{A}+N_{B}+2\leq i\leq N_{A}+N_{B}+N_{C} \end{array}$$

**Last node (*i* = *N_A_* + *N_B_* + *N_C_* + 1).** The last node is placed in the boundary between the outer jacket and air, so indoors, only convection and radiation must be considered,
(23)$$\rho_{C}S_{i,C}c_{p,C}\frac{T_{i}^{j+1}-T_{i}^{j}}{\Delta t}=k_{C}\frac{T_{i-1}^{j+1}-T_{i}^{j+1}}{\Delta x_{C}}p_{i-1}-hp_{i+1}(T_{i}^{j+1}-T_{air})-\varepsilon \sigma p_{i+1}[{\left(T_{i}^{j}\right)}^{4}-{\left(T_{air}\right)}^{4}]$$
where
(24)$$\begin{array}{l} \left\{ \begin{array}{l} S_{i,C}=\pi {\left[x_{i}\right]}^{2}-\pi {\left[x_{i}-0.5\Delta x_{C}\right]}^{2}\\ p_{i-1}=2\pi \left[x_{i}-0.5\Delta x_{C}\right]\\ p_{i+1}=2\pi R_{C} \end{array}\right.\\ \mathrm{with}\;i=N_{A}+N_{B}+N_{C}+1 \end{array}$$
where *R_C_* (m) is the outer radius of the cable, *σ* (W/(m^2^K^4^)) is the Stefan–Boltzmann constant, and *ε* = 0.85 (-) is the emissivity coefficient [36,37], whereas *h* (W/(m^2^K)) is the heat transfer coefficient. Assuming that there is no wind, i.e., the worst condition, the heat transfer coefficient due to natural convection can be calculated as [26]:(25)$$h=\frac{3.645}{\pi }\rho_{air}^{0.25}D^{-0.25}{\left(T-T_{air}\right)}^{0.25}$$

Air density *ρ**_air_* (kg/m^3^) changes with the cable elevation *H* (m), air temperature *T_air_* (K) and cable temperature *T* (K) as [26],
(26)$$\rho_{air}=\frac{1.293-1.525\cdot 10^{-4}H+6.379\cdot 10^{-9}H_{}^{2}}{1+0.00367(T_{air}+T)/2}\;$$

Finally, the TDMA coefficients of the boundary node between the outer jacket and air result in,
(27)$$\begin{array}{l} \left\{ \begin{array}{l} a_{i}=-k_{C}\frac{p_{i-1}}{\Delta x_{C}}\\ b_{i}=\frac{\rho_{C}S_{i,C}c_{p,C}}{\Delta t}+hp_{i+1}+k_{C}\frac{p_{i-1}}{\Delta x_{C}}\\ c_{i}=0\\ d_{i}=\frac{\rho_{C}S_{i,C}c_{p,C}T_{i}^{j}}{\Delta t}+hp_{i+1}T_{air}-\varepsilon \sigma p_{i+1}[{\left(T_{i}^{j}\right)}^{4}-{\left(T_{air}\right)}^{4}] \end{array}\right.\\ \mathrm{with}\;i=N_{A}+N_{B}+N_{C}+1 \end{array}$$

It is worth noting that (1)–(27) have been programmed and solved in the MATLAB^®^ environment by the authors of this work.

## 3. Experimental

### 3.1. Experimental Setup

A high-current transformer (variable output voltage 0–3 V, variable output current 0–1 kA, and 50 Hz alternating current) was used to conduct the experimental tests carried out at AMBER high-current laboratory of the Universitat Politècnica de Catalunya. To regulate the output current, the transformer has an input stage that includes an autotransformer. A loop formed by the analyzed insulated cable was directly connected to the output terminals of the transformer, as shown in Figure 8.

Figure 8b also shows the insertions made in the cable to place the thermocouples. Table 1 details the main characteristics of the insulated cable used in the experiments.

To eliminate any hot spot or heat sink, the length of the cable was 4 m, which is in agreement with the recommendations found in different international standards [38,39,40].

The current circulating through the cable loop was measured using a Rogowski coil (ACP1000 GMC-I, 1 mV/A, ±1%, DC to 10 kHz, PROsyS, Skelmersdale, UK), which provides a voltage that is linear with the electric current in the loop. The temperature in the different parts of the insulated cable (conductor-insulation boundary, insulation-jacket boundary and jacket-air boundary) was measured using low-thermal inertia welded-tip T-type thermocouples with a diameter of 0.2 mm. T-type thermocouples were selected because they are among the most accurate thermocouples, with an accuracy up to 0.5 °C. An OMEGA USB-2400 acquisition card was used to acquire the temperatures with a sampling frequency of 10 Hz. The ambient temperature was maintained at a constant value during the tests. Before the tests, it was ensured that the cable was at room temperature.

Experimental errors mainly depend on the accuracy of the Rogowski coil used to measure the current and the T-type thermocouples used to measure the temperature. Special care must be taken when performing the insertions made in the cable (conductor-insulation and insulation-jacket) to place the thermocouples, which are shown in Figure 8b. Finally, the values of different geometric conductor parameters (diameter and wall thickness of the insulation and jacket layers) as well as the physical properties of the materials (volumetric mass density, resistivity, specific heat capacity or thermal conductivity) also influence the accuracy of the simulation results.

### 3.2. Experimental Results

Experimental temperature rise tests were conducted to validate the thermal model of the conductor. These tests consist of applying current steps up to close the rated current to the cable loop during a certain time interval, during which the cable heats up in different stages. The applied current levels and durations of each current level are summarized in Table 2.

Figure 9 shows the experimental and simulated temperature profiles in the three measured points of the cable (conductor-insulation boundary, insulation-jacket boundary and jacket-air boundary) corresponding to the five current levels.

Results presented in Figure 9 show great agreement between experimental and simulated data, since the mean error is less than 1%, thus proving the suitability of the proposed cable model.

The proposed model also allows other parameters of the cable to be determined, such as the radial temperature distribution or the components of the heat balance equation, as shown in Figure 10.

It is worth noting that using an Intel^®^Core^TM^ i7-1185G7 CPU@ 3.0 GHz with 32 GB RAM (Intel, Santa Clara, CA, USA) with a time step of 10 s and nine nodes for each material, the software requires 0.7 s to run a simulation consisting of five current steps (see Figure 9) between *t* = 0 s and *t* = 8600 s.

### 3.3. Additional Experimental Results

To further validate the accuracy of the cable model, a second cable of 4 m length was tested, its characteristics are summarized in Table 3.

As with the first cable, experimental temperature rise tests were conducted to validate the proposed model of the conductor. They consist of applying five current steps to the cable loop during a certain time interval. The applied current levels and their durations are summarized in Table 4.

Figure 11 shows the experimental and simulated temperature profiles in the three measured points of the cable (conductor-insulation boundary, insulation-jacket boundary, and jacket-air boundary) for the five current levels.

Results presented in Figure 11 show great agreement between experimental and simulated data, since the mean error is also less than 1%, thus proving the accuracy of the proposed cable model.

## 4. Conclusions

This paper has presented a model for determining the current–temperature relationship between insulated and jacketed cables in air, which fully develops the physical equations governing the heat transfer problem. The model solves the transient heat transfer equations through the different layers of the cable, namely, the conductor core, the inner insulation layer, the outer insulating and protective jacket and the air surrounding the cable. To this end, the model discretizes the cable in the radial axis and applies a finite difference method approach to calculate the temperatures in all nodes of the discretized domain by considering the contiguous left-hand and right-hand side nodes. Since this approach leads to a tri-diagonal matrix, it is solved by applying the tri-diagonal matrix algorithm (TDMA). Experimental temperature rise tests performed by applying current steps of different magnitudes show the accuracy of the proposed method. The approach presented in this paper can be applied to determine the temperature rise of the cable once the applied current and ambient temperature are known, even under short-circuit conditions or under changing applied currents or ambient temperatures. The method proposed here can be easily adapted to other cable configurations.

## Figures and Tables

**Figure 1 materials-15-06814-f001:**
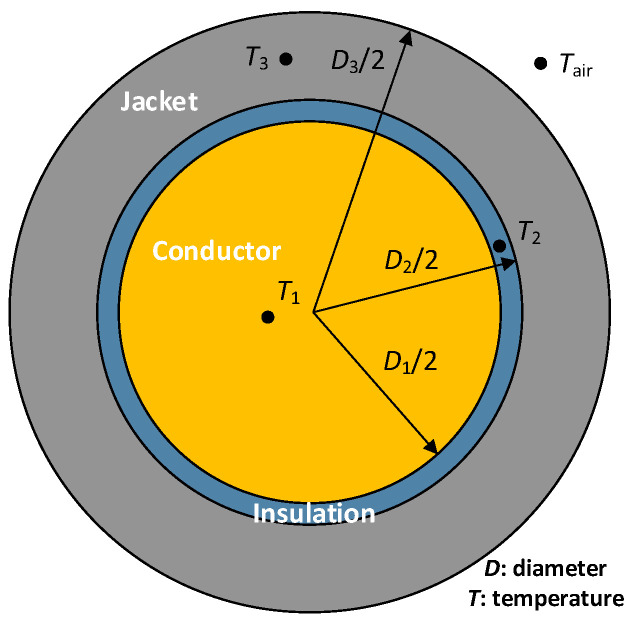
Cable layout including the copper core, the inner XLPE insulation layer and the outer PVC jacket.

**Figure 2 materials-15-06814-f002:**
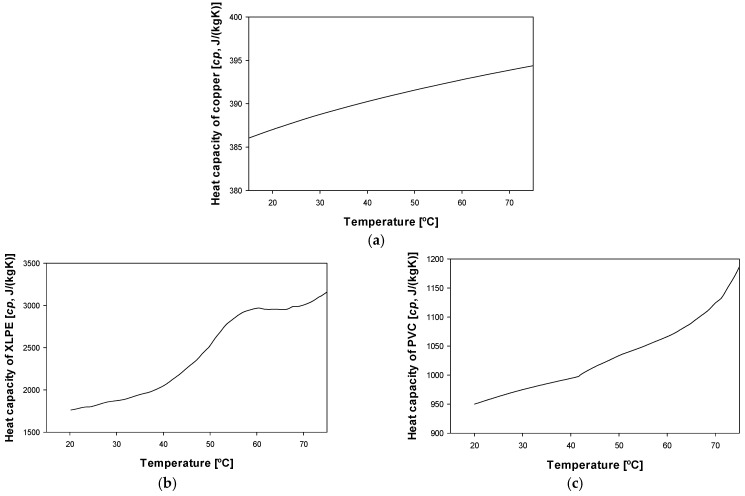
Temperature evolution of the specific heat capacity *c_p_* of the different cable materials: (**a**) Copper [28]. (**b**) XLPE [29]. (**c**) PVC [30].

**Figure 3 materials-15-06814-f003:**
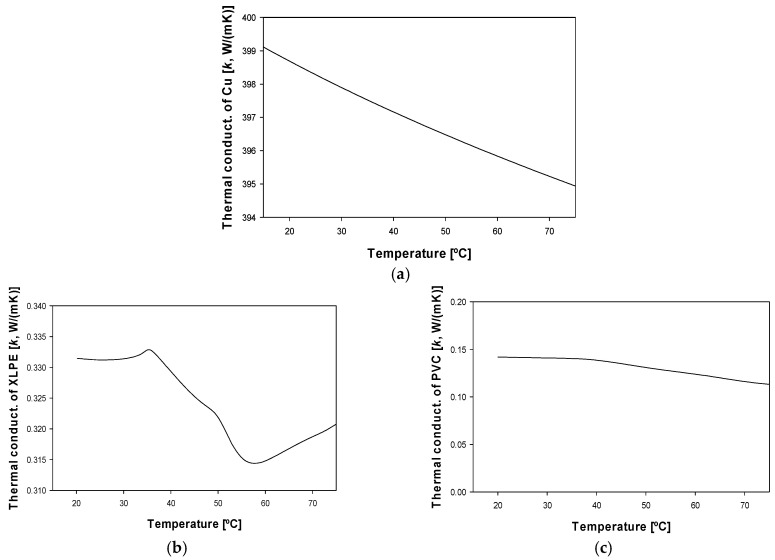
Temperature evolution of the thermal conductivity *k* of the different cable materials: (**a**) Copper [31]. (**b**) XLPE [32]. (**c**) PVC [33].

**Figure 4 materials-15-06814-f004:**
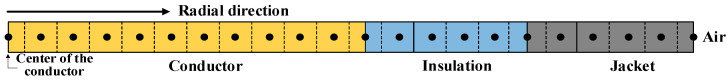
Cable discretization along the radial axis.

**Figure 5 materials-15-06814-f005:**
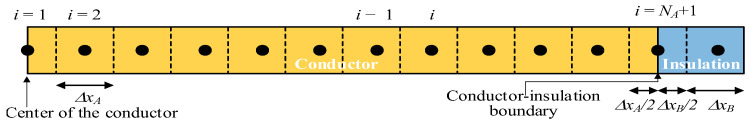
Conductor discretization.

**Figure 6 materials-15-06814-f006:**
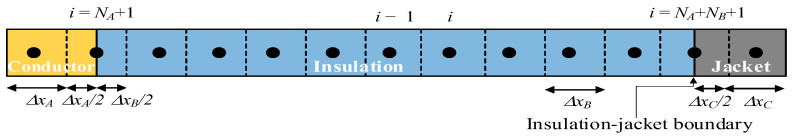
Inner insulation layer discretization.

**Figure 7 materials-15-06814-f007:**
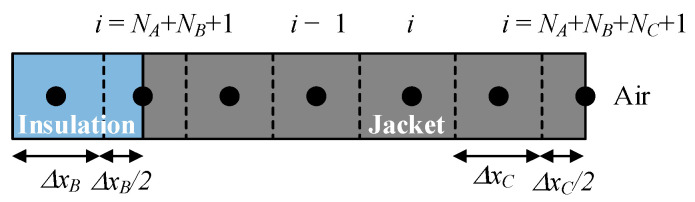
Outer jacket discretization.

**Figure 8 materials-15-06814-f008:**
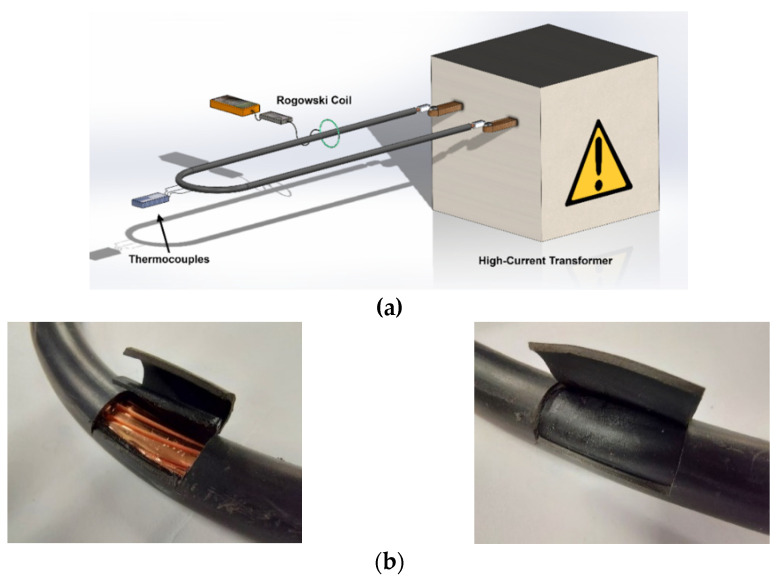
Experimental setup: (**a**) Transformer and cable loop used in the experiments. (**b**) Details of the insertions (conductor-insulation and insulation-jacket) made in the cable to place the thermocouples.

**Figure 9 materials-15-06814-f009:**
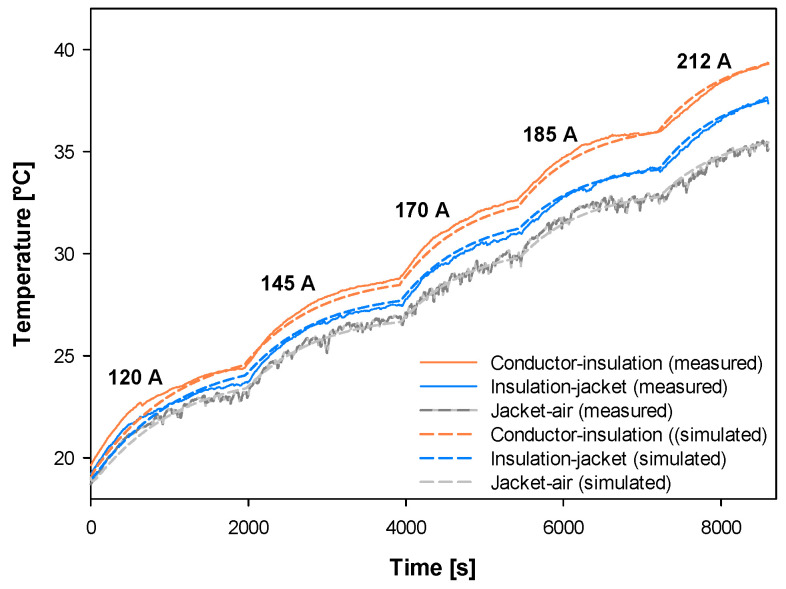
Conductor of 70 mm^2^. Experimental versus simulated temperature rise profiles in the three measured points of the cable (conductor-insulation boundary, insulation-jacket boundary and jacket-air boundary) and the relative difference between the experimental and simulated temperatures.

**Figure 10 materials-15-06814-f010:**
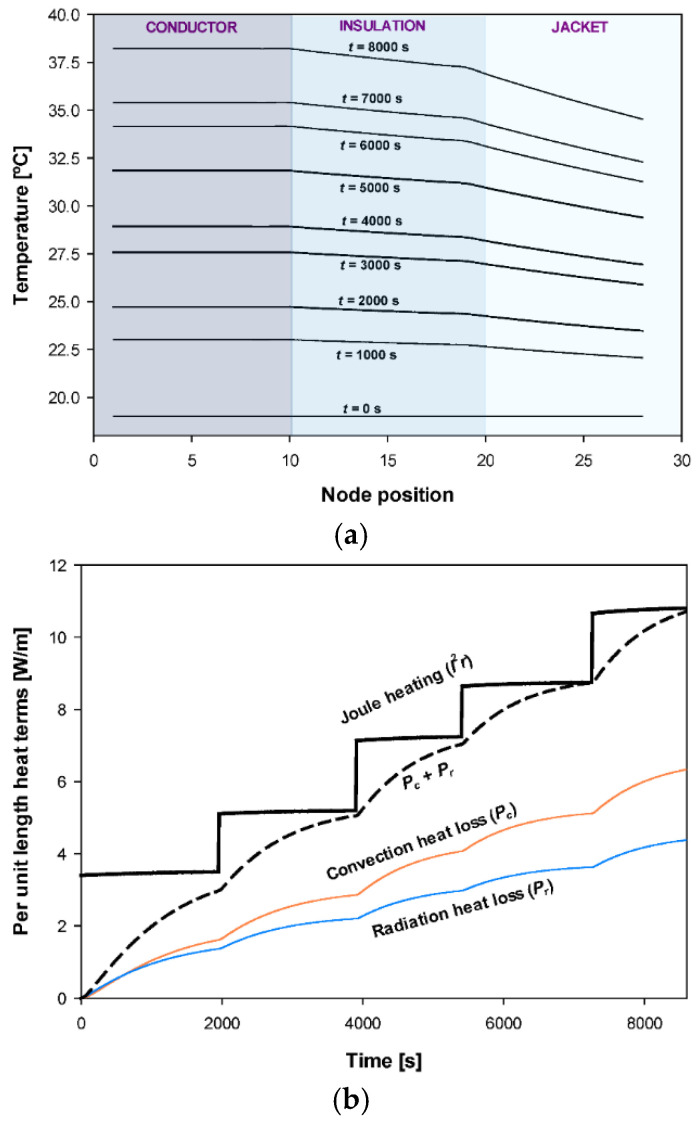
Conductor of 70 mm^2^: (**a**) Temporal evolution of the temperature along the radial axis considering 28 nodal elements. (**b**) Temporal evolution of the components of the heat balance equation.

**Figure 11 materials-15-06814-f011:**
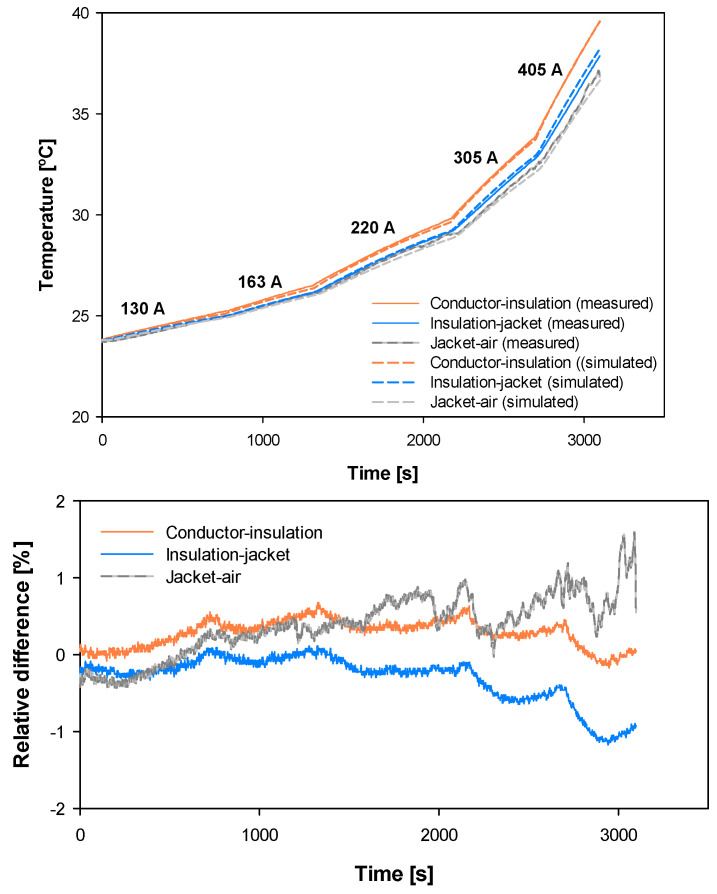
Conductor of 150 mm^2^. Experimental versus simulated temperature rise profiles in the three measured points of the cable (conductor-insulation boundary, insulation-jacket boundary and jacket-air boundary) and the relative difference between the experimental and simulated temperatures.

**Table 1 materials-15-06814-t001:** Conductor dimensions and characteristics.

Characteristic	Value
Designation	Barrinax U-1000 R2V
Rated voltage [kV_RMS_]	0.6/1.0
Max voltage [kV_RMS_]	1.2
Max continuous service temperature [°C]	90
Short circuit temperature [°C]	250
Inner insulation material	XLPE
Inner insulation wall thickness [mm]	1.1
Outer jacket material	PVC
Outer jacket wall thickness [mm]	1.5
Effective copper cross section [mm^2^]	70
Outer conductor diameter [mm]	9.5
Copper resistivity 20 °C [Ohm·m]	1.85 × 10^−8^
Temperature coefficient of resistivity [K^−1^]	0.0043
Number of strands [-]	14
Per unit length mass of the conductor [kg/m]	0.584
Ambient temperature [°C]	19

**Table 2 materials-15-06814-t002:** Conductor of 70 mm^2^. Realized temperature rise tests.

Step	Current (A_RMS_)	Duration (s)
#1	120	1950
#2	145	1950
#3	170	1500
#4	185	1850
#5	212	1350

**Table 3 materials-15-06814-t003:** Dimensions and characteristics of the second conductor.

Characteristic	Value
Designation	H07RN-F TITANEX 1 × 150
Rated voltage [kV_RMS_]	0.6/1.0
Max voltage [kV_RMS_]	1.2
Max continuous service temperature [°C]	90
Short circuit temperature [°C]	250
Inner insulation wall thickness [mm]	2.0
Outer jacket wall thickness [mm]	2.5
Effective copper cross section [mm^2^]	150
Outer conductor diameter [mm]	15
Copper resistivity 20 °C [Ohm·m]	1.85 × 10^−8^
Flexibility class	5
Per unit length mass of the conductor [kg/m]	1.74
Ambient temperature [°C]	23.5

**Table 4 materials-15-06814-t004:** Conductor of 150 mm^2^. Realized temperature rise tests.

Step	Current (A_RMS_)	Duration (s)
#1	130	766
#2	163	554
#3	220	850
#4	305	530
#5	405	400

## Data Availability

Not applicable.

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
