# Peer review of "A Model to Calculate the Current–Temperature Relationship of Insulated and Jacketed Cables"

_materials, 2022, doi:10.3390/ma15196814_

Round 1

Reviewer 1 Report

 This paper proposes a current-temperature model of the cable, and verifies it by experiments, in order to solve the problem of temperature change during power cable operation. The authors were not sufficiently trained in scientific article writing, and result in the novelty and value of this paper are not displayed clearly. The current version is very difficult to read, and I have experienced a hard time reviewing it.

Here are some of my personal recommendations. If there is a misunderstanding of what this paper does, feel free to clarify.

1.      In the introduction section, lines 26-41 only detail the gaps in each standard, but not the importance of these research gaps. An additional paragraph should be added here to emphasize the scientific significance and application value of this study.

2.      In the introduction section, after introducing the research progress, it should briefly summarize the research status and problems. Then describes the research work and significance of this paper in the next paragraph.

3.      Lines 138-139 and line 146-147 of Page 4 are two lines forming a paragraph. This writing way is very unusual. It is recommended that the authors change it to conventional writing.

4.      Figures 2 and 3 are the data of the reported literature, not this work, and it is not appropriate to occupy too much space in the paper. Here are two common methods: A. Reduce the number of graphs, which can put multiple data into one graph; B. Only show the temperature range involved in the model.

5.      The authors compare the results of the model with the experimental results to verify the accuracy of the model. The advantages of the model in this work can be better demonstrated if there is a comparison between the reported models (such as references 17-19) and the model in this paper. In addition, simulation results from COMSOL and ANSYS can be added for comparison.

6.      In the experimental verification section, in my opinion, the proposed model is an infinitely long cable. However, in Figure 8, the measured locations of the experimental setup and the cable details shown can see significant bends, which do not quite match the model conditions.

7.      Line 298-300 of Page 12, “To eliminate any hot spot or heat sink, the length of the cable was selected longer than 2 m”, what was the length of the cable in the experiment? Did the authors change the cable length to measure to rule out length interference with the experimental results?

8.      How many times was the experiment measured? Due to the accidental error of measurement, in order to eliminate interference, the average value of 5 measurements is generally taken.

9.      Experimental systematic errors, measurement method errors, etc. should be considered before comparing the measurement results with the simulated results.

Author Response

Please find our reply in the file attached

Reviewer 2 Report

The authors provide a method based on the FDM to calculate the temperature distribution for a cylindric-symmetric cable design, which allows to solve the system with one spatial coordinate as function of time. Simulations are compared with an experiment. The following points should be considered:

-          It is mentioned that the skin effect is accounted for by measuring the cable resistivity. Apparently, some effective value is obtained. In the modelling of the conductor in Section 2.2, the conductor is divided in different layers. When doing so, one could use the current density as it is calculated when including the skin effect. That means a lower value for the inside and a larger value for the outside current density. A problem then may be that a multi-physics approach is needed: the heating of the outside is stronger, resulting in a higher dissipation. This causes an increased resistance and consequently a larger skin depth. Then the current distribution changes, and so on. Would a multi-physics approach change the result significantly compared to the methodology applied by the authors?

-          Copper is a very good thermal conductor, meaning that there will hardly be a temperature variation in radial direction over the conductor. Would it not have been easier to just assign a single temperature value to the conductor due to its total Joule losses (accounting for the skin effect)? As the thermal conductivity of the conductor is about three orders of magnitude larger compared to the surrounding insulation material, a temperature drop over it seems to be negligible.

-          In equation (1) there is a term accounting for the solar radiation. This term is not further being used. The solar radiation violates the rotational symmetry of the problem description (comes from one direction) and can therefore not be included with the presented method. Therefore, consider leaving this term out of the equation.

-          Considering the index “i”, e.g., in Section 2.2: In (6) there appears x1. Is this value equal zero as in represents the centre of the conductor? E.g., in (9), the indices of two adjacent circumferences have indices “i-1” and “i+1”. This is a difference of two, where Ione would expect a difference of one (adjacent layers). Please check the indexing in all equations carefully.

-          The heat transfer in (25) includes density and ambient temperature. As it concerns a cable in open air, wind speed is as important in the heat balance as well. The authors could make a remark on this (e.g., that no wind is a kind of worst-case situation).

-          In the experimental part, what was the length of the tested cable? A too short cable makes that heat transfer in axial direction becomes also important. How did the authors verify that heat transfer along the cable to the end connections can be neglected? Further, the conductor is stranded. This means that there will be partly air between XLPE and conductor. Does this influence the heat transfer and consequently the temperature distribution?

-          In addition to the comparison in Figure 5, more details on the simulation result could be provided. What is the temperature distribution (at some time instance) in radial direction through the cable (so also inside the materials). Also, how does the radiated power flow from the cable compare to the heat losses from convection? As the maximum temperature rise during the experiment is only 20 degrees centigrade, one would expect that it can be neglected.

-          The time needed for the calculation does not look very fast. This is probably related to the very short time steps taken (0.1 s). Why this choice?

Author Response

(The authors gave the same response as above.)

Round 2

Reviewer 1 Report

The revised manuscript meet the standard of Materials. But there is an issue that needs to be paid attention to. The experimental error needs to give a specific error range. If the error in the experimental system is difficult to calculate precisely, the authors can use the experimental results of 5 times to plot the error bar in Figure 9 and Figure 11.

Author Response

REPLY: Thank you very much for your review and constructive comments that have helped to improve the quality of the manuscript. We have added an error bar in both Figure 9 and 11.